# Energy Estimation of Last Mile Electric Vehicle Routes

André Snoeck
Amazon
snoeckas@amazon.com

Aniruddha Bhargava
Amazon
baaniru@amazon.com

Daniel Merchan
Amazon
damercha@amazon.com

Josiah Davis
Amazon
davjosia@amazon.com

Julian Pachon
Amazon
pachonj@amazon.com

## ABSTRACT

Last-mile carriers increasingly incorporate electric vehicles (EVs) into their delivery fleet to achieve sustainability goals. This goal presents many challenges across multiple planning spaces including but not limited to how to plan EV routes. In this paper, we address the problem of predicting energy consumption of EVs for Last-Mile delivery routes using deep learning. We demonstrate the need to move away from thinking about range and we propose using energy as the basic unit of analysis. We share a range of deep learning solutions, beginning with a Feed Forward Neural Network (NN) and Recurrent Neural Network (RNN) and demonstrate significant accuracy improvements relative to pure physics-based and distance-based approaches. Finally, we present Route Energy Transformer (RET) a decoder-only Transformer model sized according to Chinchilla scaling laws. RET yields a +217 Basis Points (bps) improvement in Mean Absolute Percentage Error (MAPE) relative to the Feed Forward NN and a +105 bps improvement relative to the RNN.

## CCS CONCEPTS

• **Computing methodologies** → **Planning and scheduling**; **Machine learning**; • **Applied computing** → **Transportation**.

## KEYWORDS

Machine Learning, Deep Learning, Transformers, Electric Vehicles, Energy Estimation

**ACM Reference Format:**
André Snoeck, Aniruddha Bhargava, Daniel Merchan, Josiah Davis, and Julian Pachon. 2024. Energy Estimation of Last Mile Electric Vehicle Routes. In *Proceedings of August 25-29, 2024 (KDD'24 Fragile Earth Workshop)*. ACM, New York, NY, USA, 6 pages. https://doi.org/XXXXXXX.XXXXXXX

## 1 INTRODUCTION

Electrifying the last mile fleet is a key pillar of achieving sustainability goals for many last-mile carriers [7, 24, 25]. The lifecycle emissions of Electric Vehicles (EVs) are approximately 3x lower than the lifecycle emissions of Internal Combustion Engine (ICE) vehicles [11].

But from a route planning perspective, introducing EVs in last mile operations is not simply the substitution of traditional ICE vehicles with EVs. For example, EVs are constrained by their limited charging speed (measured in units of hours vs. minutes to refuel a traditional ICE vehicle). In this paper we first highlight some of the unique challenges to route planning with EVs, in particular, the need to accurately estimate energy consumption for route planing. Afterwards, we present specifics of two different deep learning models we trained to estimate energy consumption, observing significant benefits in comparison to benchmarks that rely on pure distance and physics simulations. Next, we present improvements on the baseline deep learning approaches that can be gained by increasing the model's capacity through Chinchilla scaling laws [12] and the Transformer architecture [26] which we call RET. We conclude with a discussion of additional ways to enhance energy estimation for last mile routing.

## 2 PROBLEM STATEMENT: ELECTRIFYING A LAST MILE FLEET IS ALL ABOUT ENERGY

The single most important takeaway before strategizing an EV-based last mile operation is that the currency of EV is energy consumption, not range. Concretely, it's about whether an EV with a given amount of energy in the battery can complete its route. Certainly, energy consumption is positively correlated with distance traveled. However, in last mile routing there are many other factors that influence energy consumption. For example, the amount of energy used to complete the exact same route can vary by more than 100% depending on the ambient temperature. Figure 1 visually highlights this by showing the returning State of Charge (SOC) (y-axis) for executed delivery routes with a certain distance (x-axis) across a range of ambient temperature (color) for select EVs in Europe. Observe, for example, how the returning SOC of routes at the 0.5 normalized EV distance varied between 0% and 70% which is highly correlated with temperature. Additional factors like topography, stationary time, and acceleration profile also influence energy consumption beyond the distance alone.

The realization that energy is the currency of EVs is somewhat of a mental shift from how the general public refers to the capability of EVs in terms of range. This is due to the difference between the standard consumer EVs driving cycles and those of a delivery route which consist of frequent start-and-stop on short segments, including looking for parking. Additionally, delivery routes are parked for a significant portion of the day while the driver is out delivering, so Heating, Ventilation, and Air Conditioning (HVAC)

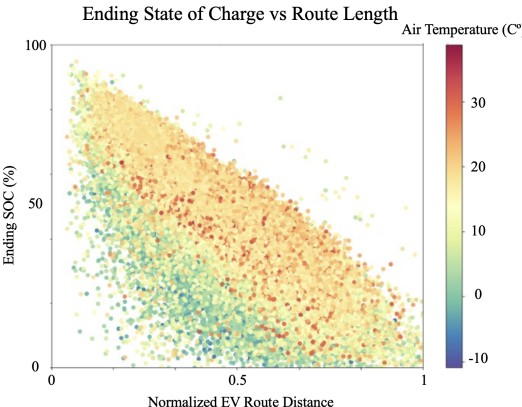

**Figure 1: Energy consumption (remaining Battery Capacity %) vs. total route length (km). Each dot is one route. Color represents ambient temperature (C°).**

consumption has a disproportional impact compared to consumers driving their EVs.

The complexity of operating EVs lies in three challenges associated to energy [13].

First, EVs are constrained by their battery capacity due to the limited charging speed, which is approximately two orders of magnitude slower than a comparable ICE vehicles. While fast-charging reduces the EV charging times significantly, it is often not a desirable option as it deteriorates the battery faster, and it imposes a significantly increased load on the power budget at the Delivery Stations (DSs).

Second, public charging infrastructure is limited and nascent. Charging locations are scattered, often less available in areas where routes have the highest risk of running out of energy, e.g., rural areas, and do not support the required scale of the fleet of last-mile carriers. Additionally, leveraging public charging on-road is impractical given the charging speed of EVs and the associated time-impact on routing.

Third, determining if a planned route is energy-feasible is not straightforward due the large set of factors influencing consumption such as weather, elevation, and others. Additionally, given the limitations on charging infrastructure and speed, there are limited recourse actions if a route turns infeasible. If an EV gets assigned an energy-infeasible route, this will lead to a poor experience for drivers, customers, and carrier.

Consequently, the available energy constraint imposes a productivity vs energy-risk trade-off. Fundamentally, adding an energy constraint to route planning deteriorates productivity and route quality metrics, as the feasible space of routes is reduced. However, not posing this constraint leads to risk of range anxiety or lost productivity (e.g., from vehicles needing to return before completing their route). This all underscores the need for accurately estimating energy consumption.

## 3 RELATED WORK

Energy estimation traditionally has been done using physics models ([17]). In these cases, signals such as time, distance, HVAC usage, air temperature, acceleration profiles, mechanical characteristics of the car (rolling resistance, weight, aerodynamic, electrical efficiency, etc.), physical characteristics of the road (slope, elevation gained, etc.) are used to compute an estimate. In this paper, we distinguish ourselves from this line of work because we have imperfect information. For example, we cannot predict traffic with full accuracy, and we can only use proxies for expected HVAC usage like predicted air temperature. Additionally, we do not have a fine-grained characteristics of the route during planning, such as exact elevation and slope.

There have been direct ML methods to predict energy, see [3, 18]. The main differences for us are the available information at planning and the latency requirements which rule out very large models. We will talk more about this in Section 6.

This type of problem of predicting a series of energy estimates can also be compared to time-series estimation framework [4, 10, 15, 22]: predicting the time-series based on the features and the history of predicted energy.

## 4 MODELING ENERGY CONSUMPTION WITH DEEP LEARNING

Predicting the energy consumption of a route accurately is paramount for energy-aware route planning. In this section, we introduce the features used to predict the energy consumption. Next, we introduce two deep learning models, the first is a Feed Forward NN which models each segment individually and the second is a RNN which models the route as a sequence of segments.

### 4.1 Features.

EV energy consumption on a delivery route is influenced by a wide range of factors with varying levels of influence. Broadly speaking, these factors can be categorized into a few major buckets: 1) route characteristics, e.g., number and sequence of stops; 2) path travelled, e.g., elevation profile, road type, and distance; 3) environment, e.g., weather conditions, and traffic; 4) vehicle, e.g., battery state of health, weight, and powertrain efficiency; and 5) additional factors like acceleration profile and HVAC usage. These categories jointly inform derived features, such as travel time, which is dependent on environmental factors (e.g., traffic), path travelled (distance), and vehicle (acceleration capability). Table 1 captures the features we included for training our deep learning models. We included features primarily based on two main criteria: feature availability and exploratory analysis of their impact to routes. For example, while package weight is important based on physics, the weight of the delivered packages on routes is insignificant relative to the weight of the vehicle. Conversely, as discussed in Section 2 temperature is an important physics-based feature that vary significantly across routes and impact energy consumption.

### 4.2 Predicting individual segments with a Fully Connected Model.

Routes consist of multiple sequential segments, i.e., travel from stop A to stop B and associated delivery operations at stop B. Our first deep learning model is a Fully Connect NN which predicts the energy consumption of segments independently. After predictions are made at the segment level, we then aggregate into route level.

**Table 1: Features included for training Deep Learning models**

| Feature | Description |
| --- | --- |
| distance | Planned travel distance (m) |
| speed_moving | Average moving speed for the segment (m/s) |
| time_stationary | Time spent parked and delivering at the end of the segment (s) |
| air_temperature | Air temperature at the start of the segment (C) |
| is_stem_int | Binary indicator if a segment is stem, i.e., traveling to or from the DS, or on-zone |
| vehicle_model | One-hot encoding of vehicle make & model |

The segment-level model is a NN with two fully-connected 32 unit layers.

## 4.3 Predicting entire routes with a Recurrent Neural Network.

While the Feed Forward NN would satisfy extremely tight latency constraints, it ignores the fact that segments in a route are not independent. They are executed by the same driver and vehicle, in weather and traffic conditions that don't vary rapidly from segment to segment. Therefore, predicting the energy consumption of a route can benefit from a route-level model, i.e., a model that considers all segments within a route jointly. We developed a route-level model that uses a RNN with a Gated Recurrent Unit (GRU)-layer (Unidirectional) to capture the correlations between segments. We consider the route as a ROUTE LENGTH x NUMBER OF FEATURES matrix. First, we use a feature embedding through a length 32 unit fully-connected layer. Next, we pass each time-step through a GRU of size 64 to keep the memory of the sequence of segments that jointly construct the whole route, followed by an output embedding with a 32 unit fully-connected layer. Lastly, we use a linear layer (a 32 x 1 matrix) to predict the output at every step. The output is a vector of energy consumption for each segment, where each prediction is influenced by the prediction for the other segments in the route. Lastly, we sum up the consumption of the individual segments.

## 5 TRAINING A COMPUTE OPTIMAL TRANSFORMER

We explore how much accuracy can be gained from using a larger model and more fully utilizing our available training data. We use a transformer architecture to test this idea as it is proven to be highly performant and scalable across a wide variety of deep learning domains beyond language including computer vision [6], speech [9] and reinforcement learning [2].

We call this model RET which is a decoder-only transformer, mostly consistent with GPT2 [21]. The primary differences between RET and GPT2 are the following: 1) we replace word embeddings with the hand-crafted feature representation of the features described previously, 2) the output dimension is 1 for the predicted energy consumption for each step in the route vs. the number of vocabulary, and 3) the cross-entropy loss is replaced with mean absolute error. But the biggest difference is that RET models are significantly smaller than GPT2 due to the smaller dataset size.

Chinchilla scaling laws [12] demonstrate how to pick the optimal model size and training data required for a given training compute budget. They use three different approaches, which generally agree. In Approach 2 the authors train dozens of models at different model parameter and data sizes, varying them across 9 different compute budgets. Then they fit a parabola on the loss for each compute budget to identify the optimal number of parameters and tokens. Finally, they demonstrate linear relationships between compute, parameter count, and data size).

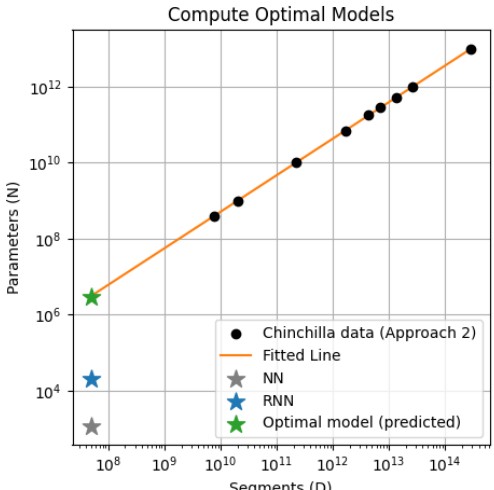

**Figure 2: Data size vs. Parameter count for Chinchilla Optimal Models. We predict that the Chinchilla optimal model size is orders of magnitude larger than the Feed Forward NN and the RNN.**

Applying these results to our problem, we quickly identified that our limiting factor is not training compute, but rather data quantity. Using their published data [12] we fit a linear model to interpolate the optimal number of model parameters $N$, for the amount of data $D$ we have, yielding:

$$\log_{10}(N) = 0.51 \log_{10}(D) + 0.0617$$

Using imputation, we determine the optimal model size to be 3M parameters. This is visualized in Figure 2 along with the Feed Forward NN and RNN.

In Chinchilla [12], optimality is defined as getting the most "bang for buck": maximizing model accuracy for the given compute budget. However, for any optimization system the model's inference compute is also critical. Hence we also train two smaller models (Table 2), to evaluate whether we can get some of the same benefits of the larger transformer with reduced inference time.

## 6 MODELING RESULTS

We present a holistic evaluation of the modeling effort, including overall MAPE, MAPE on a subset of Cold and Hot routes, and inference speed.

André Snoeck, Aniruddha Bhargava, Daniel Merchan, Josiah Davis, and Julian Pachon

**Table 2: RET variants. The 20k parameter version (RET-20k) is chosen as a point of reference to the RNN model which has the same number of parameters and the 300k version is chosen as a middle-ground approach which combines some benefit of the larger model with lower latency.**

| Model | Blocks | Dimension |
|---|---|---|
| RET-20k | 1 | 32 |
| RET-300k | 3 | 96 |
| RET-3M | 6 | 192 |

## 6.1 Accuracy

We train models on 48M segments across North America and Europe. We evaluate the model using Mean Absolute Percentage Error (MAPE) on 5M segments. Additionally, we evaluate the models on a hold-out subset of several hundred routes that are particularly hot ($\geq 35°C$) and are particularly cold ($\leq 0°C$). We compare accuracy with two baselines. The first baseline (Distance) is a simple linear model which predicts energy consumption from the distance of the route as a proxy for the 'range' concept. The second baseline (Physics) is an internal physics-based simulation. We report accuracy improvements relative to the Fully Connected NN in terms of basis points (bps) of improvement in MAPE. Comparing the NN model to the distance and physics baselines in Table 3, we observe a sizable benefit of +1117 bps and +260 bps respectively. Moving to the RNN, we observe an additional benefit of +112 bps on top of the NN. This likely due to a combination two things: modeling the sequence as opposed to independent segments, as well as an overall larger model size.

Comparing to the NN, we observe a +217 bps performance increase in MAPE from training the Chinchilla optimal RET-3M. Comparing to the RNN model we observe an additional +105 bps improvement. We also compare results on hold-out set for particularly hot and cold routes. The hot and cold routes are particularly important because energy usage tends to be higher due to increased HVAC usage. Additionally, cold weather may impact battery capacity, increasing risk without proper estimates. For particularly hot routes, this improvement grows astonishingly to +753 bps. Additionally, RET-3M performance is also demonstrated for cold routes where we observe an improvement of 167 bps in MAPE %. This is is noteworthy because cold routes were relatively lacking in the training data.

## 6.2 Inference Speed

In order to realize the benefits of the larger models more compute time is required. Even though these models are microscopic in relation to modern Deep Learning models used for Language and Vision, computational performance is noteworthy due to the following observation. As important as energy estimation is, it is only a single input into computationally intensive downstream optimization programs. For example, as a part of routing optimization for a single delivery station, the energy estimation model might need to be evaluate several orders of magnitude more routes than we end up needing. Therefore, being cognizant of inference time is crucial to ensure the model is feasible to use.

**Table 3: MAPE results from model experimentation. Each row represents the bps improvement relative to NN for one model/training. Overall results are reported as well as results for the subset hot and cold routes.**

| Model | Relative MAPE% | Relative MAPE% Cold $\leq$ 0°C | Relative MAPE% Hot $\geq$ 35°C |
|---|---|---|---|
| Distance | -1117.8 | -893.2 | -1971.1 |
| Physics | -260.5 | -276.5 | 500.4 |
| NN | 0.0 | 0.0 | 0.0 |
| RNN | 111.8 | 129.2 | 563.6 |
| RET-20k | 123.3 | 79.9 | 671.1 |
| RET-300k | 194.9 | 143.9 | 760.0 |
| RET-3M | 216.9 | 167.2 | 753.6 |

**Table 4: Inference Speed Comparison. Each row represents one model/training, and inference speed is reported on single host for 10k routes.**

| Model | CPU (s) | GPU (s) |
|---|---|---|
| NN | 0.107 | 0.242 |
| RNN | 2.422 | 0.406 |
| RET-20k | 0.857 | 0.013 |
| RET-300k | 7.133 | 0.671 |
| RET-3M | 27.373 | 2.572 |

To quantify the relative speed between the models, we measured the inference time on a single host: c6i.32xlarge for CPU and p3.2xlarge for GPU. We used the same python framework we used for training the models (Keras for NN and RNN and PyTorch for RET). We report the average forward pass time across 5 forward passes, after giving the models two warm-ups. Using this setup, we make two observations. First, using the GPU speeds up runtime by 10x for the larger two models. Second, RET-20k is much faster than RNN which is also 20k parameters, meaningful because RET-20k and RNN models have a similar overall performance on MAPE. Finally, while there is sub-linear growth in inference speed as the model size grows, RET-3M is still 4x slower (CPU and GPU) than RET-300k. This is meaningful since RET-300k approaches the same MAPE overall and for cold routes, and in fact has a slight edge for hot routes.

## 7 FINAL REMARKS

In this paper, we demonstrated the need to estimate energy consumption for EV aware route planning. We also demonstrated the benefit of a range of Deep Learning-based solutions to predict energy consumption, including a compute-optimal Transformer model, RET.

*Beyond route planning.* In this paper, we have argued there exists a route quality vs route-risk trade-off due to the introduction of limited battery capacity constraints in route planning. Since energy consumption cannot be predicted with 100% accuracy, an explicit decision has to be made about how conservative we define the routing constraint. However, route planning is one of the last steps

in a typical last-mile planning process. Upstream decisions such as vehicle design, network topology, and vehicle scheduling can create conditions where the battery capacity constraint only has limited impact on the routing outcome. For example, in the presence of a mixed battery-capacity fleet, smartly assigning delivery vehicles with larger battery capacities to areas with higher consumption routes limits the impact of the EV fleet on route quality. Additionally, route planning is not the final step in a last-mile planning process. Real-time, on-the-road support to keep drivers informed about their remaining SOC and the anticipated feasibility of their route, in light of weather or traffic changes, ensures we are able to adjust routes as needed to minimize the impact on successful completion of the route. For each of these planning processes in the end-to-end last mile workflow, having accurate estimates of the required energy of a set of (expected) routes is critical, and the applicability and impact of models as introduced in this paper span beyond traditional routing problems. This ensures that, in addition to the societal benefits of EVs in achieving sustainability goals, EVs are also the cost-optimal optimal choice for a last-mile carrier.

*Future research.* As part of future research, we strive to increase model accuracy and reliability. One path forward is to explore what we can leverage from the physics model. We know that the energy consumption is determined by physical laws and can be computed with 100% accuracy in a full-knowledge setting, i.e., knowing the exact values for all relevant features such as air temperature, speed, and elevation. One option would be to start with the physics predictions and build a model that predicts the delta between the physics model and the actual energy used explicitly forcing the model to learn signals not currently captured by the physics model.

Another area for improvement is explicitly modeling the uncertainty. There is a rich area of previous work that we could apply to this problem [8]. We can look at conformal prediction [1, 16, 23], quantile regression [14] and tools in deep learning with uncertainty (TensorFlow Probability, Bayesian Neural Networks, Epistemic Neural Networks etc.) see [5, 19, 20]. Most methods for uncertainty rely on ensembles, estimating a distribution and then doing Monte Carlo sampling or creating a larger network to predict the distribution. All these increase the inference time. This is an active area for future work.

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

## A  GLOSSARY

**DS** Delivery Station

**EV** Electric Vehicle

**GRU** Gated Recurrent Unit

**HVAC** Heating, Ventilation, and Air Conditioning

**ICE** Internal Combustion Engine

**MAPE** Mean Absolute Percentage Error

**ML** Machine Learning

**NN** Neural Network

**RNN** Recurrent Neural Network

**RET** Route Energy Transformer

André Snoeck, Aniruddha Bhargava, Daniel Merchan, Josiah Davis, and Julian Pachon

**SOC**  State of Charge