# OpenReview forum: "Energy Estimation of Last Mile Electric Vehicle Routes"
_KDD.org/2024/Workshop/Fragile_Earth — Fragile Earth FullPresentation_

### Official Review · Reviewer_qREq · 2024-07-12
**Well written paper that addresses an important and challenging problem.**

**Rating:** 9
**Confidence:** 4

**Review:**

This paper illustrates how deep neural network based architectures can help understand the unique challenges of last mile delivery related problems when EVs are being used. The paper is well written and sparks a different thinking in terms of how the challenges of delivering with EVs are different than the consumer grade EVs. Those include more frequent stops, ambient temperature changes, the weight (due to the carried packages) as well as the fast charging challenges in terms of the battery strain and infrastructural impacts.

The proposed approach focuses on (instead of discussing the range) if a given route can be completed with the existing charge level of a vehicle. It uses a novel Route Energy Transformer (RET) model which improves the MAPE substantially compared to the baseline approaches. Authors also conducted an analysis for the inference performance which can be an important aspect considering each delivery route (per day / per deliveries planned) may be different and these consumption inferences may need to be run fast enough to create new optimal (or achievable) routes every time for each EV.

Overall, I think the paper is well written and clearly shows how a state-of-the-art NN architecture can help mitigate challenges on a specific problem. Therefore, I will be happy to see it presented in the workshop.

---

### Official Review · Reviewer_ti9U · 2024-07-13
**A well-motivated paper and well-executed paper on an important challenge**

**Rating:** 8
**Confidence:** 4

**Review:**

This paper address the problem of the energy consumption and feasibility of the last mile delivery plans for Amazon as they move to electric vehicles (EV). The paper shows how the well designed combination of state-of-the-art deep learning architectures, including feed forward NN, RNN and Transformer can lead to significant improvement in the prediction accuracy in the specific problem setting. The paper additionally points out that it is necessary to focus on the energy consumption rather than the range in this new scenario. The paper is well motivated, well written and targets a compelling challenge grounded in a unique but significant real world problem having to do with delivery routing for addressing sustainable development goals.

---

### Official Review · Reviewer_Pr62 · 2024-07-15
**Review of "Energy Estimation of Amazon's Last Mile Electric Vehicle Routes"**

**Rating:** 8
**Confidence:** 4

**Review:**

The paper focuses on predicting energy consumption of delivery EV.  Different ML models, including feed forward NN, RNN and a specialized transformer, are used for such predictions. Their accuracy and the speed of inference are compared in empirical evaluation.

Strengths:
(1) The paper is well-written and engaging. It nicely introduces the motivation, the problem setup, and the design choice of the methods.
(2) I believe this paper study an important problem in sustainable societal development.
(3) I like Section 5 of the paper that reasons about the trade-off between ML performance and model sizes.

Weaknesses:
There are a few clarity issues:
(1) It needs a bit more description of the dataset used. What are “ca. 48M”, “ca. 5M”? These were not introduced in the paper.
(2) Improvement is measured in bps but the measurement bps is not explained or defined.

---

### Decision · Program_Chairs · 2024-07-24

Accept (Full Presentation)